# Biological Potential of Fruit and Leaves of Strawberry Tree (*Arbutus unedo* L.) from Croatia

**DOI:** 10.3390/molecules25215102

**Published:** 2020-11-03

**Authors:** Jana Šic Žlabur, Sandro Bogdanović, Sandra Voća, Martina Skendrović Babojelić

**Affiliations:** 1Department of Agricultural Technology, Storage and Transport, Faculty of Agriculture, University of Zagreb, Svetošimunska cesta 25, 10000 Zagreb, Croatia; jszlabur@agr.hr (J.Š.Ž.); svoca@agr.hr (S.V.); 2Department of Agricultural Botany, Faculty of Agriculture, University of Zagreb, Svetošimunska cesta 25, 10000 Zagreb, Croatia; 3Centre of Excellence for Biodiversity and Molecular Plant Breeding, Svetošimunska cesta 25, 10000 Zagreb, Croatia; 4Department of Pomology, Faculty of Agriculture, University of Zagreb, Svetošimunska cesta 25, 10000 Zagreb, Croatia; mskendrovic@agr.hr

**Keywords:** *Arbutus unedo* L., bioactive compounds, chemical composition, micro-location, antioxidant capacity

## Abstract

The strawberry tree fruit and leaf are a rich source of minerals, easily digestible sugars, dietary fibers, vitamins (especially vitamin C) and many bioactive compounds of significant functional value. Due to their favorable chemical composition, fruits have recently become increasingly popular in consumption. The aim of this study was to determine the physical-chemical composition, content of bioactive compounds, and also the antioxidant capacity of the fruit and leaves of wild strawberry tree populations among the Adriatic coast in Croatia, as well as to investigate the influence of location on the content of specific bioactive compounds. According to the obtained results, both fruit and leaves are pronouncedly high in vitamin C content, the average value for fruits amounted to 224.21 mg/100 g FW, while that for leaves amounted to 138.08 mg/100 g FW. Additionally, significantly high values of total polyphenolic compounds were recorded both in fruits (average value of 637.94 mg GAE/100 g FW) and especially in leaves (average value of 2157.01 mg GAE/100 g FW). Several pigments from different categories were determined in the fruit depending on fruit maturity, including: total anthocyanins, β-carotene and lycopene; while in leaves chlorophylls and carotenoids. Given the high content of different bioactive compounds high values of antioxidant capacity were determined (the average value for fruits was 2269.96 µmol TE/kg and for leaves, 2237.16 µmol TE/kg). Location strongly influenced the physical-chemical composition and also the content of specialized metabolites; populations collected from southern areas (central and south Dalmatia) of the Adriatic coast tended to have higher amounts of vitamin C, total phenols, total anthocyanins and β-carotene.

## 1. Introduction

The strawberry tree (*Arbutus unedo* L., fam. Ericaceae) is a neglected fruit species in Croatia and its potential for cultivation and consumption has not been sufficiently utilized. Namely, the fruit and leaves of the strawberry tree are of significant functional importance and many studies have proven high contents of minerals, easily digestible sugars, dietary fibers, vitamins (especially vitamin C) and also many other bioactive compounds with significant antioxidant activity in strawberry fruit and leaves [1,2]. Due to strawberries’ rich chemical composition, they have recently become increasingly popular for consumption, primarily due to their favorable nutritional composition, and therefore significant medicinal properties: antiseptic, diuretic, laxative, anti-inflammatory action, prevention of diabetes and others [3,4,5]. The beneficial effect of the strawberry tree on health has been recognized by consumers, which is why almost all parts of the plant have been applied in traditional medicines [6]. Due to the high content of carbohydrates, fruits ferment quickly and are not as highly valued as for fresh consumption. Additionally, given the abovementioned, from a health perspective, excessive consumption of fruits is not recommended [5]. Strawberry tree leaves are also a nutritionally valuable raw material and various scientific studies have proven the high content of vitamin C and specific phytochemicals, bioactive compounds such as polyphenolic compounds (arbutin), terpenoids, essential oils, tocopherols, etc., present in strawberries [7,8]. Based on the above, the use of leaves, as well as leaf products, has suggested a number of biological benefits including significant antioxidant activity [9], antibacterial and antifungal effects [10] and anti-inflammatory action of strawberry tree leaves [11]. Therefore, the strawberry tree is an undervalued fruit species, with different possible commercial uses from the food industry (jams, jellies, alcoholic beverages) to ornamental, pharmaceutical and chemical industries [12,13].

The strawberry tree is an evergreen fruit species whose natural populations are widely distributed in the eu-Mediterranean region. In Europe it grows throughout the Mediterranean basin, from Portugal to Turkey. In Croatia, the Strawberry tree is a wild fruit species distributed from Istria to Dubrovnik, where it forms typical Mediterranean forests that are characterized by high temperatures and dry and sunny climatic conditions [14]. As a heliophilous species, it grows alongside holm oak, evergreen Mediterranean forests and macchias [1,6,14,15], with a very narrow ecological niche, growing on different soil types and with different amounts of precipitation. Because of this, the natural populations of strawberry trees can have a significant impact on the content of nutritional compounds in the fruit and leaves of the plant, especially in the Mediterranean part of Croatia where the northern regions (Istria and Kvarner islands) are less warm and sunny than the southern regions (central and south Dalmatia, along with the islands) [16].

The aim of this study was to determine the chemical composition and content of bioactive compounds in the fruit and leaves of wild strawberry tree populations, collected from natural habitats in the Mediterranean part of Croatia, characterized by significantly different pedological and climatic conditions.

## 2. Results and Discussion

### 2.1. Physical-Chemical Composition of Strawberry Tree Fruit

The physical-chemical composition of strawberry tree fruits from different locations are presented in Table 1. All analyzed parameters strongly differ with significant statistical difference (*p* ≤ 0.0001), given the varied location. Dry matter content (DM, %) ranges from 28.53–44.89%. Despite a fairly wide range of determined DM values, in all analyzed fruit samples high values were determined suggesting that the fruits of the strawberry tree represent a high quality nutritional raw material. Additionally, compared to the other literature data, DM results obtained in this study are lower, suggesting that climatic conditions and location significantly influence the DM content [17,18]. The lowest dry matter content was determined for fruits from location one, while the highest was from location six. Fruits of strawberry tree populations collected from geographically closer areas (micro-locations), tended to have similar DM values. For example, in fruits from the Kvarner area (locations three, four and five) an average DM value of 35% was determined; in fruits from central Dalmatia (locations 8, 9 and 10) an average DM value of 33.5% was obtained, and fruits from south Dalmatia (locations 11, 12, 13, 16 and 17) yielded an average DM value of 32%, with some recorded deviations in each researched area. The obtained results suggest that strawberry tree fruits collected from the more northern areas of the Adriatic coast, tended to have higher DM values compared to the populations from southern areas. The total soluble solids content (TSS, %) results are presented in Table 1, ranging from 21.67 to 29.8% with an average TSS, for fruits from all locations within the Adriatic coast, of 24.24%. Regardless of the location, recorded TSS values in all samples can be classified as relatively high in comparison with other fruit species. Namely, the fruits of strawberry tree species are characteristic for their high sugar content, with average values determined from the literature data suggesting values of between 15 to 30% [18,19,20,21]. Comparing the results of TSS from the literature data, it can be concluded that the fruits anaylzed in this research are in accordance with previous findings. Analyzing the results of TSS values depending on location, can be confirmed that the lowest TSS value was determined in the fruits from location 16 (south Dalmatia area), while the highest TSS value was in the fruits from location seven (Kvarner area). The obtained results for TSS values in relation to the location of strawberry tree populations, are in agreement with the trend of the results for DM content, suggesting that in populations from northern areas of the Adriatic coast higher TSS contents are determined. The Istrian area had an average of 26.5% TSS, the Kvarner area had an average of 24.5%, central Dalmatia demonstrated an average of 23% and south Dalmatia had an average of 23%, with some recorded deviations in each area. The possible reason for the recorded trend, both for DM and TSS content, could be due ecological factors which strongly differ with respect to the stated locations; southern areas of the Adriatic coast tend to have higher average air temperatures (~17 °C) and lower levels of precipitation (~831 mm) per year compared to the northern areas (Istria: ~15 °C, 896 mm; Kvarner: ~16 °C, 1216.5 mm; central Dalmatia: ~16 °C, 868 mm) [16].

Based on the results obtained for total acid content (TA, %) of strawberry tree fruits (Table 1), it can be seen that there is a significant distribution of the results in the range from 0.39 to 1.18%. The average determined TA content of fruits from all researched locations within the Adriatic coast amounts 0.74%. The previous literature data state values of TA content in the range of 0.66–1.59% [6], or even higher values of between 1.51–3.45% [22]. The lowest TA value was recorded in location eight, while the highest TA value was in location six. Taking into account the locations of the sampled populations, an average TA value of 0.7% was determined in fruits from the Istrian area (locations one and two), an average TA value of 0.59% in fruits from the Kvarner area and 0.82% in fruits from south Dalmatia. TA values recorded in fruits from the central Dalmatia area deviate more significantly between each micro-location; for example in fruits from location eight, the lowest TA content was recorded, while in fruits from location 10, among some of the highest determined contents (0.86%) were recorded, with these two locations belonging to the same area (central Dalmatia). Determined pH values (Table 1) of strawberry tree fruits also differ within the researched locations, with an average value of 3.32. Minimal standard deviation from the mean was recorded for the pH values of strawberry tree fruits from different locations, which indicates that all strawberry tree fruit samples have a fairly uniform pH value regardless of location. The obtained pH values were similar to those found in the strawberry tree fruits from Spain [3,23] with an average value of 3.43.

### 2.2. Physical-Chemical Composition of Strawberry Tree Leaves

Since strawberry tree leaves are edible in their fresh state, analysis of leaf chemical composition is essential. Additionally, a relatively small amount of scientific studies have analyzed the physical-chemical composition of strawberry tree leaves, so further research is necessary [2]. The physical-chemical compositions of strawberry tree leaves collected from different locations are shown in Table 1. According to the obtained results, high significant statistical difference (*p* ≤ 0.0001) was determined for all analyzed chemical parameters depending on location. In general, all Strawberry tree leaf samples, regardless of the population location, show a high content of dry matter (DM, %), ranging from 50.93 to 91.9%. A wide range of DM contents, with respect to the location of analyzed populations, was noted. The lowest DM content was determined in samples from location 11, while the highest content was in samples from location six. On average, in leaves from populations collected in the Istrian area, the DM content was 59.61%, from the Kvarner area the average DM value was 85.5%, from central Dalmatia it was 73.15% and from south Dalmatia it was 70.09%. A similar trend of lower DM contents of strawberry tree leaves in southern locations, compared to the northern locations, was also found, as was the case for DM contents of the strawberry fruits. Total acid content (TA, %), also significantly differed between strawberry tree leaves in relation to the location, in the range of 0.71 to 1.93%. Since the strawberry tree leaf is a poor source of total soluble solids (TSS), primarily sugars, the TSS content was not determined in samples from wild strawberry tree populations collected from different locations. Additionally, in the scientific literature there is a lack of available data regarding TSS content and specific sugar content of strawberry tree leaves. The lowest TA content was determined in leaves from strawberry tree populations from location two, while the highest was in leaves from location six. The average TA content value of strawberry leaf samples from all analyzed locations amounts 1.22%. pH values (Table 1) range from 3.89 (location three) to 5.35 (location nine) and suggest a medium-high acidity of the samples. The average pH value of the strawberry tree leaves amounts 5.06, regardless of the location. As in the case of TSS content, by reviewing the scientific literature, a lack of available data was also identified for TA content and pH values of the leaves, so further studies into the chemical composition of strawberry leaves is of great importance.

### 2.3. Bioactive Compounds Content of Strawberry Tree Fruits and Leaves

Bioactive compounds are important molecules in terms of human health, first of all because of their significant antioxidant activity [24]. A number of phytochemicals can be classified as bioactive compounds, such as: vitamins (vitamin C), minerals (selenium), pigments (chlorophylls, carotenoids), glucosinolates, polyphenols and others [25]. Additionally, lots of scientific studies have proven the positive correlation between high content of bioactive compounds and high antioxidant activity [26,27]. Both strawberry tree fruits and leaves are rich sources of bioactive compounds [8,18,28], thus suggesting that these raw materials are of great importance for human nutrition, and health in general.

The bioactive compound content and antioxidant activity of strawberry tree fruits and leaves from different locations are shown in Table 2. For all analyzed bioactive compound contents—both for fruits and leaves—high statistical significance (*p* ≤ 0.0001) was determined considering the location of strawberry tree populations. Vitamin C content of fruits varied from 107.63 (location 16) to 402.41 mg/100 g FW (location 10), suggesting a relatively wide spread of the results in relation to location. Considering the researched areas within the Adriatic coast, the following vitamin C contents were determined: in fruits from the Istrian area, the average vitamin C value was 224.1 mg/100 g FW; in fruits from the Kvarner area, the average vitamin C value was 223.41 mg/100 g FW; in fruits from central Dalmatia, the average vitamin C value was 316.87 mg/100 g FW; and in fruits from south Dalmatia the average vitamin C value was 189.97 mg/100 g FW. With regard to environmental factors (temperature, precipitation and number of sunny days) specific to each individual area researched in this study, such dispersion of vitamin C results is expected. It is important to emphasize that one of the main factors for vitamin C synthesis and accumulation in fruit is the rate of photosynthesis since one of the main products of photosynthesis, glucose, is a precursor in the synthesis of vitamin C in plant tissues [29]. So, higher vitamin C content is expected in the areas with more sunny days such as locations in central and south Dalmatia, with the exception of the Istrian area, compared to the areas with lower annual values of sunny days. Regardless of the location, in fruits of all analyzed populations, high vitamin C contents were determined, which is in agreement with other studies [13,30], or even higher compared to the values obtained from research from Turkey [31]. Besides from the strawberry tree fruit, the leaves of the researched species are specifically high in vitamin C content. Vitamin C content in strawberry tree leaves ranges from 61.61 (location two) to 333.83 mg/100 g FW (location 10). The highest vitamin C content was determined both in the fruits and leaves from the population harvested from location 10. Average vitamin C values for strawberry tree leaves from the Istrian area was 66.11 mg/100 g FW, the Kvarner area was 161.77 mg/100 g FW, central Dalmatia was 200.16 mg/100 g FW and from south Dalmatia was 134.84 mg/100 g FW. The same trend of vitamin C content in leaves regarding the researched area was observed as for the strawberry tree fruits; in general leaves collected from populations in central and south Dalmatia have higher vitamin C contents compared to those from northern areas (the Istrian area), with the exception of the Kvarner area which includes islands of specific climatic conditions (similar to central and south Dalmatia).

Polyphenol compounds are important secondary plant metabolites, responsible for the plant’s response to stress conditions [32,33]. In human cells, polyphenols are inhibitors of oxidation processes due to their ability to neutralize free radicals [34]. Polyphenols are present in all plant organs and tissues and are mainly concentrated in the fruit skin and leaves [35]. Numerous studies emphasize that strawberry tree fruits are a rich source of polyphenolic compounds, especially from the phenolic acids and flavonoids groups [31,36,37]. Results of total phenol (TPC), total non-flavonoid (TNFC) and total flavonoid (TFC) contents of strawberry tree fruits and leaves are presented in Table 2. Extremely high values for all analyzed polyphenol compounds were determined both in fruits and leaves of the strawberry tree. TPC content of fruits varied from 479.62 (location 12) to 850.02 mg GAE/100 g FW (location six), TNFC from 233.95 (location one) to 410.74 mg GAE/100 g FW (location 10) and TFC from 235.39 (location 12) to 466.88 mg GAE/100 g FW (location 17). Location had strong impact on all analyzed polyphenol compounds, since significant variability of results was determined regarding each studied area. On average, the TPC value was 357.06 mg GAE/100 g FW in fruits from the Istrian area, 474.92 mg GAE/100 g FW in fruits from the Kvarner area, 632.23 mg GAE/100 g FW in fruits from central Dalmatia and 718.59 mg GAE/100 g FW in fruits from South Dalmatia. In general, strawberry tree fruits from the populations in southern areas (central and south Dalmatia) of the Adriatic coast tended to have higher TPC values compared to the northern areas (Kvarner and Istrian areas). A similar trend was also determined for TNFC and TFC contents, with some more pronounced deviations. The determined trend of TPC fruit content can be explained by environmental factors, primarily temperature and precipitation. As previously mentioned, polyphenols are plant secondary metabolites, plant molecules which are direct responses to stressful conditions to which the plant may be exposed. High air temperature and lack of water (drought stress) are strong signals of stress to which the plant responds by increasing synthesis of defense mechanism compounds—secondary metabolites such as polyphenols. In general, it can be concluded that the fruits collected from studied areas characterized by higher air temperatures and lower amounts of precipitation (central and south Dalmatia) tended to produce and accumulate higher amounts of polyphenols [38,39].

According to the results of the polyphenol profiles of strawberry tree leaves (Table 2), leaves are also a rich source of these phytochemicals [1,31]. This is of a great importance due to the further possible usage of strawberry tree leaves, especially as a raw materials for the extraction of valuable phenols, which than can then be used in the production of various products for number of phyto-therapeutic purposes. The average TPC content in strawberry tree leaves was 2157.01 mg GAE/100 g FW; TNFC was 1778.27 mg GAE/100 g FW and TFC was 378.74 mg GAE/100 g, regardless of the particular location. The location of individual strawberry tree populations significantly affected on the content of all studied polyphenolic compounds, with the same being true for plant leaves. For the specific areas within the Adriatic coast, the following TPC contents of leaves have been determined: for the Istrian area—an average value of 2091.97 mg GAE/100 g, for the Kvarner area—an average value of 2042.90 mg GAE/100 g, for central Dalmatia—an average value of 2242.91 mg GAE/100 g and for south Dalmatia—an average value of 2528.42 mg GAE/100 g. As was found for the fruits, plant leaves also followed the trend of increasing TPC content from the northern to the southern areas of the Adriatic coast, with the highest TPC values determined in the leaves of populations from southern Dalmatia. In general, the lowest TPC content in strawberry plant leaves (1577.4 mg GAE/100 g) was determined for location 12, while the highest value (2693.81 mg GAE/100 g) was from location 17. It is interesting that both the lowest and the highest TPC contents were measured in the same area (south Dalmatia), which proves that the micro-location, i.e., the specific ecological factors within each general location, significantly affect the content and profiling of polyphenol compounds. Variability of TNFC and TFC values in leaves was also recorded for researched location and area: the TNFC value was on average 1556.57 mg GAE/100 g FW and the TFC value was on average 535.72 mg GAE/100 g FW in the Istrian area; the average TNFC value was 1648.31 mg GAE/100 g FW and the average TFC value was 394.75 mg GAE/100 g FW in the Kvarner area, the average TNFC value was 1813.98 mg GAE/100 g FW and the average TFC value was 428.94 mg GAE/100 g FW in central Dalmatia, while in south Dalmatia the average TNFC value was 2173.28 mg GAE/100 g FW and the average TFC value was 535.72 mg GAE/100 g FW.

### 2.4. Bioactive Colored Compounds of Strawberry Tree Fruit and Leaves

Plant pigments are natural colorants which give fruits, leaves and other plant parts their specific coloration. Besides coloration, specific groups of plant pigments, also have different roles in the plant. For example, chlorophylls are crucial for the photosynthesis process, while carotenoids have a protective role, act as photo-protectors [40] and are potential antioxidants during plant stress [41]. Except for their role in plants, plant pigments are important for humans too. Specifically, many pigments that are naturally present in plants are efficient antioxidants and have a number of potential benefits for human health. Anthocyanins, have been proven to have a strong antioxidant effect, along with β-carotene from the carotenoid group [40]. Strawberry tree fruits are also unique due to their long period of maturation, whereby fruits from the same plant may show different compositions of individual pigments [6]. In this study, several plant pigments both in the fruits (Table 3) and leaves (Table 4) were analyzed. In strawberry tree fruit, three types of plant pigment were identified: anthocyanins, β-carotene and lycopene. The content of all the analyzed plant pigments differ significantly (*p* ≤ 0.0001) dependent on the location. In fruits from locations 1, 6, 10 and 14, total anthocyanin content (TAC) was not determined, suggesting that the micro-location might have a significant role in their accumulation. From the Istrian area, fruit TAC levels were only determined in fruits from location two with an average value of 3.08 mg/kg, from the Kvarner area the average value was 7.69 mg/kg, from central Dalmatia the average value was 8.1 mg/kg and from south Dalmatia average values were 12.23 mg/kg. The obtained results suggest that fruits from the southern part of the Adriatic coast tended to have higher TAC values compared to those from the northern area, despite the highly determined variability of results regarding each location. In general, the highest TAC (24.19 mg/kg) was determined in fruits from location 11 (the area of south Dalmatia), and it can be assumed that the climatic conditions of south Dalmatia (the number of sunny days and temperature) are conducive to faster fruit ripening, and thus, the accumulation of anthocyanins. The other two analyzed pigments are the most common ones from the group of carotenoids and are responsible for giving fruits their specific orange (β-carotene)/red (lycopene) coloration. β-carotene content in fruits collected from different locations ranged from 50.07 (location 11) to 560.89 µg/100 g FW (location 18). With respect to the studied locations, the content of β-carotene differs significantly between fruit populations. From observing specific areas within the Adriatic coast, the following average values from each area were determined: the Istrian area had an average β-carotene value of 109.72 µg/100 g FW, the Kvarner area had an average value of 280.82 µg/100 g FW, central Dalmatia had an average value of 191.3 µg/100 g FW and south Dalmatia had an average value of 299.15 µg/100 g FW. The increasing trend of β-carotene content from northern to southern areas was also determined as for the TAC, with the exception of the Kvarner area in which slightly higher values of β-carotene were determined compared to central Dalmatia. Lycopene content was determined in only a few strawberry tree fruit samples, depending on the location, while in general, in all samples low lycopene content was identified. The highest lycopene content (1.14 mg/g) was determined in fruits from location six and the lowest content (0.21 mg/g) in fruits from locations 7, 14, 16. Strawberry tree fruits are not known for having high lycopene content, as evidenced by other scientific studies [1].

The composition of the pigments in the strawberry tree leaves significantly differs from the fruit, with chlorophyll as the most common, as expected (Table 4). All analyzed pigments from leaves are significantly different (*p* ≤ 0.0001) depending on the location. Chlorophyll a content (Chlor_a) ranged from 0.19 (location six) to 1.13 mg/g (location eight), while chlorophyll b content (Chlor_b) ranged from 0.21 (location six) to 1.24 mg/g (location eight) and the total chlorophyll content (TCh) ranged from 0.40 (location six) to 2.37 mg/g (location eight). From the analyzed samples it can be pointed out that leaves from location six had the highest content of all the researched chlorophylls, while samples from location eight had the lowest chlorophyll content. The results of all analyzed chlorophyll compounds are strongly dispersed considering the investigated areas and also the location (micro-location) of specific area. In general, in leaves from the Istrian area an average TCh value of 1.42 mg/g was determined, in leaves from the Kvarner area an average value of 0.81 mg/g was determined, from central Dalmatia an average value of 1.43 mg/g was found and in leaves from south Dalmatia average value of 1.23 mg/g was noted. All plant tissues which contain chlorophyll pigments also contain carotenoids, as is confirmed by the results of this research. In strawberry tree leaves, the total carotenoid content (TCA) ranged from 0.06 to 0.27 mg/g, while the lowest TCA was determined in leaves from location three and the highest in leaves from location 14. Considering the researched areas within the Adriatic coast, leaves collected from strawberry tree populations from the Istrian area had an average TCA value of 0.17 mg/g, those from the Kvarner area had an average value of 0.1 mg/g, those from central Dalmatia had an average value of 0.14 mg/g and those from south Dalmatia had an average value of 0.195 mg/g. As for the results of chlorophyll content, the results for TCA are strongly dispersed considering the investigated areas and also the location (micro-location) of specific area.

### 2.5. Antioxidant Capacity of Strawberry Tree Fruit and Leaves

The determined values for antioxidant capacity, both of strawberry tree fruits and leaves, are relatively high (Figure 1), suggesting that the analyzed plant material is of great importance from a functional point of view. Oxidation processes are extremely undesirable in human cells and in the cells of all living beings for that matter, leading to cell decay and accelerating the aging process. Some biomolecules have a significant ability to slow down oxidative processes by inhibiting the action of free radicals, showing great antioxidant activity and thus having a great impact on human health by preventing the number of chronic diseases; from cardiovascular, neurodegenerative to cancer [42]. One such class of biomolecules, with strong antioxidant activity are vitamins (especially vitamin C), numerous compounds from the group of pigments (especially carotenoids), while the most numerous—polyphenolic compounds (especially anthocyanins, flavonoids and others). All of these phytochemicals were identified in the fruit and leaf samples of strawberry tree species suggesting the importance of this species in nutrition. Antioxidant capacity (Ant_cap) both of Strawberry tree fruits and leaves significantly differed depending on the location. In strawberry tree fruits Ant_cap was in the range of 2251.65 to 2284.14 µmol TE/kg, while in leaves it was between 2205.17 to 2297.09 µmol TE/kg. Other scientific data also prove the high antioxidant activity of edible parts of strawberry tree species [37], suggesting that the obtained results of this research are in agreement with other published data.

## 3. Materials and Methods

### 3.1. Plant Material

The research was conducted throughout 2017 by sampling the fruits and leaves from wild strawberry tree populations at 18 different locations, shown in detail in Table 5 and in Figure 2. Geographical coordinates for each locality were recorded using Garmin vista e-Trex GPS (Garmin International, Inc., Hampshire, UK). Collected plant material (herbarium specimens) was digitalized and deposited in the ZAGR Herbarium of the Faculty of Agriculture, University of Zagreb. Herbarium specimens are accessible through the Virtual Herbarium ZAGR (http://herbarium.agr.hr/).

Leaf and fruit samples were collected from three randomly selected strawberry tree populations (each tree from the same location represented one repetition). Only healthy and ripe fruits and healthy leaves were collected for the intended laboratory analyses. The collected plant material (fruits and leaves) was transported to the laboratory of the Department of Agricultural Technology, Storage and Transport at the University of Zagreb, Faculty of Agriculture, for further chemical analysis.

### 3.2. Determination of Physical-Chemical Composition of Strawberry Tree Fruit and Leaves

For the purposes of the physical-chemical composition determination of the strawberry tree fruit and leaves, standard methods were used: dry matter content (DM, %) by drying to constant mass at 105 °C according to [44], total acidity (TA, %) by potentiometric titration according to [44], pH value by digital pH meter (Mettler Toledo, SevenMulti, Switzerland) [44], and also in the fruit, total soluble solids content (%) by digital refractometer (Mettler Toledo, Refracto 30PX, SevenMulti, Greifensee, Switzerland).

### 3.3. Determination of Bioactive Compounds Content and Antioxidant Capacity

*Vitamin C.* Vitamin C content (mg/100 g fresh weight) was determined by titration with 2,6-dichloroindophenol (DKF) according to [45]. Vitamin C was isolated from the plant material (fruit and leaf) by homogenizing the 10 g ± 0.01 of fresh plant material with 100 mL of 2% (*v*/*v*) oxalic acid. Prepared solution was filtered through Whatman filter paper and the filtrate was obtained in a volume of 10 mL. This was then titrated with fresh prepared 2,6-dichloroindophenol until the appearance of a pink coloration. The final vitamin C content was calculated according to Equation (1) and expressed as mg/100 g fresh weight.
(1)$$\mathrm{Vitamin}\;C=\frac{V\;\left(\mathrm{DKF}\right)\times F}{D}\times 100$$
where is: V (DKF)—volume of DKF (mL); F—factor of DKF; D—sample mass used for titration

Total polyphenolic compounds. The total phenolic content (TPC), including flavonoids (TFC) and non-flavonoids (TNFC) values were determined spectrophotometrically (Shimadzu, UV 1650 PC) by a method based on a colored reaction that phenols develop with Folin–Ciocalteu reagent measured at 750 nm [46]. The extraction of polyphenolic compounds was made according to the following procedure: 10 g ± 0.01 of fresh plant material was weighed into an Erlenmeyer flask and the first 40 mL of 80% EtOH (*v*/*v*) was added; the prepared sample was heated to boiling point and additionally heated for 10 min with reflux. After 10 min, sample was filtered through Whatman filter paper in a volumetric flask volume of 100 mL. After filtration, the rest of the sample was transferred in the Erlenmeyer flask and an additional 50 mL of 80% EtOH (*v*/*v*) was added, and the procedure with reflux was repeated for 10 min. The sample was filtrated in the same volumetric flask; the filtrates were combined and the flask was filled to the mark with 80% EtOH (*v*/*v*). Such prepared extract was used for reaction with Folin–Ciocalteu reagent. In a volumetric flask a volume of 50 mL 0.5 mL of ethanolic extract was added and the following chemicals were added: 30 mL od distilled water (dH_2_O), 2.5 mL of prepared Folin–Ciocalteu reagent (1:2 with dH_2_O) and 7.5 mL of saturated sodium carbonate solution (Na_2_CO_3_); the flask was filled to the mark with dH_2_O and the prepared sample was allowed to stand at room temperature for 2 h with intermittent shaking. For the purpose of TNFC content determination, the same ethanolic extracts prepared for TPC determination were used, while TNFC separation was made by the following procedure: 10 mL of ethanolic extract was added in the volumetric flask volume of 25 mL, then 5 mL of HCl (1:4, *v*/*v*) and 5 mL of formaldehyde was added. The prepared samples were blown with nitrogen (N_2_) and allowed to stand for 24 h at room temperature in a dark place. After 24 h, the same Folin–Ciocalteu reaction as for the TPC was conducted. The absorbance of the blue color both in the TPC and TNFC reaction was measured spectrophotometrically at 750 nm with distilled water as a blank probe. Gallic acid and catechin were used as an external standard and the concentration of TPC and TNFC content was expressed as mg GAE/100 g fresh weight. TFC content was expressed mathematically as the difference between total phenols and non-flavonoids.

Bioactive colored compounds. The following pigment compounds in the strawberry tree fruit following were analyzed:

Total anthocyanin content spectrophotometrically at 520 nm by the bisulphite bleaching method according to [46]: 2 g ± 0.01 of fresh fruit material was weighed into cuvette volume of 50 mL, 2 mL of 0.1% HCl (diluted with 96% EtOH) and 40 mL of 2% HCl (*v*/*v*) was added. Prepared samples were centrifuged 10 min at 4500 rpm. After centrifugation, 10 mL of supernatant was separated into one test tube and 10 mL into another test tube. In first test tube 4 mL od dH_2_O was added (blank probe), while in second 4 mL of fresh prepared solution of 15% sodium bisulphite (NaHSO_3_, *v*/*v*). Such prepared samples was allowed to stand in test tubes for 15 min, while after, absorbances (A1 for blank probe and A2 for reaction with bisulphite) were measured spectrophotometrically at 520 nm with 2% HCl (*v/v*) as a blank probe. The final anthocyanin content (TAC) was calculated by the difference of absorbance (A1–A2) multiplicated with the molar mass of the most common anthocyanin, and expressed as mg/kg.

β-carotene content was determined by the method described in [47] with some minor modifications in the isolation procedure. In a glass laboratory test tube 0.5 g ± 0.01 of plant material was weighed and a total volume of 15 mL of petroleum ether (40–70 °C): acetone (1:1) was added three times. After each addition of solvent, samples were homogenized by a laboratory homogenizer (IKA, UltraTurax T-18, Staufen, Germany). To remove the acetone from the samples, elution with distilled water was conducted, while for the purpose of water removal, samples were passed through anhydrous sodium sulphate. Final separation of β-carotene from the singled carotenoid layer was conducted by column chromatography with a specific adsorbent. Eluted β-carotene was transferred into a volumetric flask volume of 10 mL, the concentration was read spectrophotometrically at 450 nm with petroleum ether (40–70 °C) as a blank probe, while final results were recalculated and expressed as μg/100 g fresh weight.

Lycopene content was determined according to method described in [48] and adapted to the fruit samples. The homogenized fruit sample was weighed (0.3 g ± 0.01) into a glass beaker and following chemicals were added: 5 mL of 0.05% BHT dissolved in acetone (p.a.), 5 mL of ethanol (96%) and 10 mL of hexane (p.a.). The samples were incubated in a water bath at 4 °C for 15 min, after which 3 mL of distilled water was added, and the resulting suspension was left in a water bath at 4 °C for 5 min. The samples were then incubated for 5 min at room temperature with intermittent shaking. After incubation on the surface of the suspension, the hexane (colored, lycopene) layer was separated and carefully transferred into a cuvette. The absorbance was measured spectrophotometrically at 503 nm with hexane as a blank. The final results were expressed as mg/g.

Total chlorophyll and carotenoid content. Analysis included determination of chlorophyll a, chlorophyll b, total chlorophyll and total carotenoids by the method described in [49] and [50]. Pigments were isolated from plant material as follows: in a glass laboratory test tube 0.2 g ± 0.01 of fresh plant material was weighed and a total volume of 15 mL of acetone (p.a.) was added three times. After each addition of acetone, samples were homogenized with a laboratory homogenizer (IKA, UltraTurax T-18, Staufen, Germany). The final solution was filtered and transferred into a volumetric flask volume of 25 mL. The absorbance was measured spectrophotometrically (Shimadzu UV 1650 PC, Kyoto, Japan) at 662, 644, and 440 nm with acetone as a blank probe. To obtain the results of pigment content based on the complained absorbance Holm–Wettstein equations were used (2). The final results for the pigment contents were expressed in mg/g.
(2)$$\mathrm{chlorophyll}\;a\;=9.784\times A_{662}-0.990\times A_{644}\;[\mathrm{mg}/L]$$
(3)$$\mathrm{chlorophyll}\;b\;=21.426\times A_{644}-4.65\times A_{662}\;[\mathrm{mg}/L]$$
(4)$$\mathrm{total}\;\mathrm{chlorophyll}\;=5.134\times A_{662}+20.436\times A_{644}\;[\mathrm{mg}/L]$$
(5)$$\mathrm{total}\;\mathrm{carotenoids}\;=4.695\times A_{440}-0.268\times \left(\mathrm{total}\;\mathrm{chlorophlls}\right)\;[\mathrm{mg}/L]$$

### 3.4. Antioxidant Capacity Determination Greifensee

For the purposes of antioxidant capacity determination ethanolic extracts, obtained from total phenol determination, were used. Antioxidant capacity was determined spectrophotometrically using an ABTS radical cation (2,2′-azinobis (3-ethylbenzothiazoline-6-sulfonic acid)) at 734 nm according to method described in [51]. As an antioxidant standard, Trolox (6-hydroxy-2,5,7,8-tetramethylchroman-2-carboxylic acid, Sigma-Aldrich, St. Louis, MO, USA) was used. Trolox was prepared as stock standard and also appropriate dilutions for the purpose of calibration curve creation was prepared. The 5 mL of ABTS solution (7 mM) and 88 mL of potassium persulfate (140 mM) solution were mixed and allowed to stand in the dark at room temperature for 16 h for obtaining an ABTS radical (ABTS·1). On the day of analysis, 1% ABTS·1 solution (in 96% ethanol) was prepared. A total of 160 µL of extract was directly injected into the cuvette and mixed with 2 mL of stable ABTS·1. The final results were expressed as mmol TE/L.

### 3.5. Statistical Analysis

A sample of 20 fruits and 20 leaves were included in three repetitions (a total of 60 fruits and 60 leaves for each location). Laboratory analyses of chemical composition and bioactive compounds for each sample were performed in triplicate. According to a randomized block design (with three replicates), to determine the significance of differences within the examined factor ANOVA and Duncan’s multiple range tests were performed using statistical software [52]. Mean values were compared by an LSD test where *p* = 1% was considered the statistical level of significance. In addition to the results, the tables show different letters that indicate significant statistical differences between the varied treatments at *p* ≤ 0.0001. The average deviation of the results from the mean value for each investigated parameter with the values of standard deviation is also presented.

## 4. Conclusions

Based on the obtained results, it can be concluded that both strawberry tree fruit and leaves are nutritionally important sources, since in analyzed plant material samples high contents of bioactive compounds, especially vitamin C and phenols, but also different plant pigments: β-carotene, anthocyanins and lycopene in fruits and chlorophylls in leaves were identified. Based on the valuable nutritional composition and high level of bioactive compounds, high values of antioxidant capacity, both for fruits and leaves were determined. In strawberry tree populations collected from southern areas (central and south Dalmatia) of the Adriatic coast, higher contents of important nutrients such as vitamin C, total phenols, anthocyanins and β-carotene were found. Additionally, populations collected on the Mediterranean islands off the southern Adriatic coast tended to accumulate more phytochemicals in fruits and leaves. Therefore, it can be concluded that location, but also micro-location significantly influenced the nutritional quality of the edible parts—fruits and leaves—of the strawberry tree. Additionally, further studies of individual bioactive compounds from categories of polyphenols and pigments by more precise analytical methods (HPLC) are necessary in order to provide more detailed information of the potential use and quality of strawberry tree fruit and leaves. According to the results, fruits and leaves of the strawberry tree’s natural populations show significant biological potential, rich nutritional composition, and thus, a functional value ofgreat importance for the human diet.

## Figures and Tables

**Figure 1 molecules-25-05102-f001:**
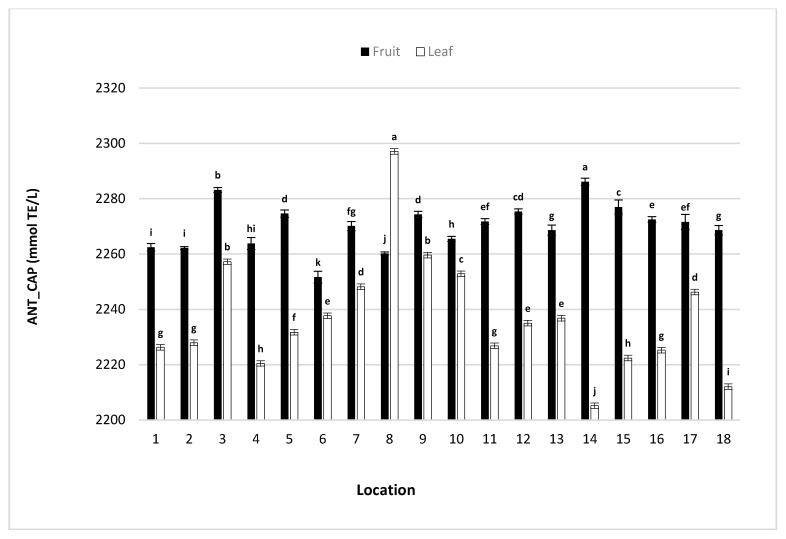
Antioxidant capacity (mmol TE/L) of strawberry tree fruit and leaves collected from different locations within the Adriatic coast. Different letters (a–k) indicate significant differences between means at *p* ≤ 0.0001.

**Figure 2 molecules-25-05102-f002:**
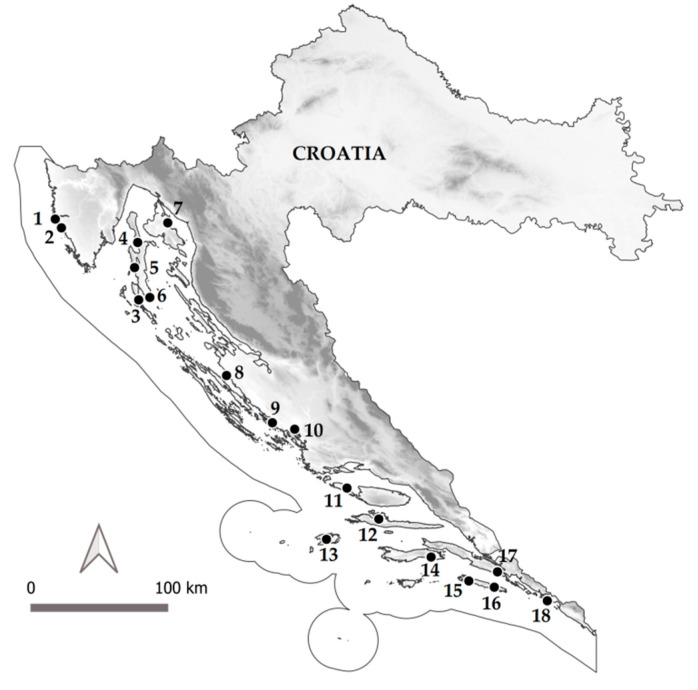
Geographical position of sampled wild strawberry tree (*Arbutus unedo* L.) populations in Croatia.

**Table 1 molecules-25-05102-t001:** Physical-chemical composition parameters of Strawberry tree fruit and leaf.

Location	DM (%)	TSS (%)	TA (%)	pH
**Fruit**				
Rovinj I (1)	28.85 ± 0.23 ^l^	26.03 ± 0.64 ^bc^	0.69 ± 0.006 ^ef^	3.32 ± 0.025 ^def^
Rovinj II (2)	36.78d ± 0.53 ^e^	26.97 ± 0.40 ^b^	0.71 ± 0.025 ^de^	3.43 ± 0.015 ^a^
Lošinj (3)	32.85 ± 1.87 ^ghij^	24.50 ± 0.85 ^def^	0.58 ± 0.05 ^h^	3.36 ± 0.02b ^cd^
Cres I (4)	34.96 ± 0.21 ^efg^	23.37 ± 0.06 ^fg^	0.60 ± 0.01 ^h^	3.32 ± 0.06 ^def^
Cres II (5)	35.85 ± 0.77 ^ef^	25.53 ± 0.38 ^cd^	0.61 ± 0.02 ^h^	3.31 ± 0.015 ^def^
Cres III (6)	44.89 ± 2.92 ^a^	25.10 ± 0.10 ^cde^	1.18 ± 0.01 ^a^	3.22 ± 0.11 ^g^
Krk (7)	38.02 ± 0.27 ^cd^	29.80 ± 1.04 ^a^	0.95 ± 0.02 ^b^	3.19 ± 0.001 ^g^
Zadar (8)	34.48 ± 0.08 ^fgh^	22.03 ± 0.29 ^hi^	0.40 ± 0.01 ^i^	3.44 ± 0.06 ^a^
Tisno (9)	31.77 ± 0.44 ^ijk^	23.03 ± 1.11 ^gh^	0.72 ± 0.01 ^de^	3.30 ± 0.03 ^def^
Šibenik (10)	34.20 ± 0.39 ^fgh^	24.63 ± 0.40 ^def^	0.86 ± 0.02 ^c^	3.40 ± 0.12 ^abc^
Šolta (11)	30.78 ± 0.47 ^jkl^	21.80 ± 0.70 ^hi^	0.70 ± 0.02 ^def^	3.34 ± 0.02 ^cde^
Hvar (12)	33.56 ± 0.37 ^ghi^	24.57 ± 1.65 ^def^	0.73 ± 0.006 ^d^	3.32 ± 0.01 ^def^
Vis (13)	32.54 ± 0.39 ^hij^	23.57 ± 1.15 ^fg^	0.70 ± 0.01 ^de^	3.28 ± 0.01 ^f^
Korčula (14)	39.30 ± 0.94 ^bc^	23.50 ± 0.57 ^fg^	0.64 ± 0.01 ^g^	3.42 ± 0.02 ^ab^
Mljet I (15)	40.53 ± 0.51 ^b^	24.00 ± 1.39 ^efg^	0.87 ± 0.02 ^c^	3.29 ± 0.02 ^ef^
Mljet II (16)	34.70 ± 3.57 ^efgh^	21.67 ± 0.21 ^i^	0.67 ± 0.02 ^fg^	3.29 ± 0.02 ^ef^
Pelješac (17)	29.97 ± 1.71 ^kl^	21.73 ± 0.49 ^hi^	0.85 ± 0.01 ^c^	3.28 ± 0.03 ^f^
Dubrovnik (18)	38.21 ± 0.05 ^cd^	24.50 ± 0.70 ^def^	0.93 ± 0.02 ^b^	3.31 ± 0.01 ^def^
ANOVA	*p* ≤ 0.0001	*p* ≤ 0.0001	*p* ≤ 0.0001	*p* ≤ 0.0001
**Leaf**				
Rovinj I (1)	61.43 ± 0.18 ^n^	ND	1.10 ± 0.02 ^gh^	5.12 ± 0.07 ^bcde^
Rovinj II (2)	57.78 ± 0.21 ^o^	ND	0.71 ± 0.02 ^j^	5.08 ± 0.03 ^cde^
Lošinj (3)	90.04 ± 0.11 ^c^	ND	1.44 ± 0.006 ^cd^	3.89 ± 0.47 ^g^
Cres I (4)	90.66 ± 0.06 ^b^	ND	1.39 ± 0.09 ^de^	4.75 ± 0.025 ^f^
Cres II (5)	75.71 ± 0.58 ^i^	ND	0.90 ± 0.02 ^i^	5.29 ± 0.04 ^ab^
Cres III (6)	91.90 ± 0.01 ^a^	ND	1.93 ± 0.02 ^a^	5.05 ± 0.02 ^de^
Krk (7)	82.44 ± 0.18 ^f^	ND	1.53 ± 0.04 ^b^	5.25 ± 0.02 ^abc^
Zadar (8)	57.53 ± 0.045 ^o^	ND	0.75 ± 0.05 ^j^	5.29 ± 0.12 ^ab^
Tisno (9)	77.61 ±0.22 ^h^	ND	1.11 ± 0.05 ^gh^	5.35 ± 0.015 ^a^
Šibenik (10)	84.32 ± 0.19 ^d^	ND	1.49 ± 0.03 ^bc^	5.19 ± 0.001 ^abcde^
Šolta (11)	50.93 ± 0.21 ^p^	ND	0.75 ± 0.006 ^j^	5.15 ± 0.006 ^bcde^
Hvar (12)	83.59 ± 0.21 ^e^	ND	1.56 ± 0.09 ^b^	5.02 ± 0.025 ^e^
Vis (13)	64.09 ± 0.25 ^m^	ND	1.17 ± 0.03 ^g^	4.68 ± 0.035 ^f^
Korčula (14)	75.07 ± 0.17 ^j^	ND	1.28 ± 0.03 ^f^	5.16 ± 0.006 ^abcde^
Mljet I (15)	66.07± 0.79 ^l^	ND	1.06 ± 0.02 ^h^	5.19 ± 0.32 ^abdce^
Mljet II (16)	70.39 ± 0.36 ^k^	ND	1.06 ± 0.07 ^h^	5.23 ± 0.05 ^abcd^
Pelješac (17)	80.45 ± 0.18 ^g^	ND	1.31 ± 0.09 ^ef^	5.12 ± 0.05 ^bcde^
Dubrovnik (18)	82.54 ± 0.15 ^f^	ND	1.40 ± 0.03 ^d^	5.16 ± 0.035 ^abcde^
ANOVA	*p* ≤ 0.0001	*p* ≤ 0.0001	*p* ≤ 0.0001	*p* ≤ 0.0001

Different letters (a–m) indicate significant differences between means at *p* ≤ 0.0001; DM—total dry matter content; TSS—total soluble solids content; TA—total acid content; ND—not determined.

**Table 2 molecules-25-05102-t002:** Bioactive compounds content of strawberry tree fruits and leaves from different locations.

Location	Vitamin C (mg/100 g FW)	TPC (mg GAE/100 g FW)	TNFC (mg GAE/100 g FW)	TFC (mg GAE/100 g FW)
**Fruit**				
Rovinj I (1)	245.34 ± 2.78 ^f^	573.89 ± 1.15 ^m^	233.95 ± 1.14 ^l^	339.94 ± 1.81 ^h^
Rovinj II (2)	202.86 ± 3.21 ^i^	657.37 ± 4.13 ^f^	283.18 ± 0.49 ^g^	374.19 ± 2.83 ^e^
Lošinj (3)	185.76 ± 3.56 ^jk^	605.39 ± 1.03 ^k^	261.72 ± 0.85 ^ij^	343.67 ± 1.52 ^gh^
Cres I (4)	124.27 ± 1.75 ^n^	649.21 ± 1.88 ^g^	291.05 ± 1.17 ^f^	358.16 ± 1.84 ^f^
Cres II (5)	190.59 ± 1.98 ^j^	553.41 ± 1.87 ^o^	258.81 ± 0.51 ^j^	294.59 ± 1.38 ^l^
Cres III (6)	326.65 ± 1.07 ^b^	850.02 ± 1.05 ^a^	394.01 ± 2.07 ^b^	457.67 ± 0.13 ^b^
Krk (7)	289.80 ± 1.28 ^d^	666.39 ± 1.77 ^e^	266.19 ± 0.84 ^i^	400.21 ± 1.11 ^d^
Zadar (8)	239.96 ± 1.76 ^g^	637.99 ± 3.17 ^h^	293.58 ± 0.49 ^ef^	344.42 ± 2.7 ^gh^
Tisno (9)	308.25 ± 2.75 ^c^	523.65 ± 1.34 ^p^	281.01 ± 0.57 ^g^	242.65 ± 1.09 ^m^
Šibenik (10)	402.41 ± 3.25 ^a^	735.04 ± 1.93 ^c^	410.75 ± 0.33 ^a^	324.32 ± 1.55 ^j^
Šolta (11)	183.23 ± 1.21 ^kl^	627.78 ± 1.91 ^i^	296.33 ± 0.47 ^e^	331.44 ± 1.66 ^i^
Hvar (12)	139.15 ± 5.23 ^m^	479.62 ± 2.42 ^q^	244.22 ± 1.25 ^k^	235.39 ± 1.21 ^n^
Vis (13)	179.42 ± 0.42 ^l^	557.65 ± 1.62 ^n^	263.57 ± 2.53 ^i^	293.74 ± 1.04 ^l^
Korčula (14)	190.08 ± 2.08 ^j^	622.24 ± 1.39 ^j^	274.51 ± 1.96 ^h^	347.73 ± 1.74 ^g^
Mljet I (15)	238.92 ± 1.41 ^g^	600.40 ± 0.65 ^l^	280.11 ± 1.03 ^g^	320.29 ± 1.08 ^jk^
Mljet II (16)	107.63 ± 1.47 ^o^	621.84 ± 0.86 ^j^	306.93 ± 1.75 ^d^	314.91 ± 2.11 ^k^
Pelješac (17)	252.28 ± 1.31 ^e^	813.54 ± 2.68 ^b^	346.65 ± 1.8 ^c^	466.88 ± 2.27 ^a^
Dubrovnik (18)	229.07 ± 2.22 ^h^	707.26 ± 1.03 ^d^	258.51 ± 0.96 ^j^	448.76±1.99 ^c^
ANOVA	*p* ≤ 0.0001	*p* ≤ 0.0001	*p* ≤ 0.0001	*p* ≤ 0.0001
**Leaf**				
Rovinj I (1)	70.61 ± 1.75 ^lm^	2084.02 ± 3.31 ^l^	1462.98 ± 0.87 ^p^	621.04 ± 2.46 ^d^
Rovinj II (2)	61.61 ± 2.44 ^m^	2099.91 ± 1.15 ^k^	1650.15 ± 0.06 ^j^	449.75 ± 1.14 ^h^
Lošinj (3)	113.63 ± 1.23 ^f^	2280.53 ± 2.04 ^e^	1604.16 ± 0.89 ^k^	676.37 ± 2.14 ^b^
Cres I (4)	156.37 ± 2.12 ^e^	2032.42 ± 2.07 ^m^	1865.99 ± 0.06 ^h^	166.42 ± 2.07 ^n^
Cres II (5)	84.44 ± 2.26 ^j^	1868.98 ± 2.09 ^p^	1717.85 ± 0.58 ^i^	151.13 ± 2.26 ^o^
Cres III (6)	187.54 ± 1.33 ^d^	2148.34 ± 2.65 ^h^	1501.65 ± 0.93 ^o^	646.68 ± 2.90 ^c^
Krk (7)	266.87 ± 1.61 ^b^	1884.25 ± 0.82 ^o^	1551.09 ± 1.08 ^m^	333.15 ± 1.52 ^j^
Zadar (8)	81.01 ± 2.56 ^jk^	2341.44 ± 1.09 ^d^	1561.85 ± 0.68 ^l^	779.62 ± 1.26 ^a^
Tisno (9)	185.63 ± 2.61 ^d^	2180.49 ± 1.39 ^g^	1908.47 ± 2.77 ^g^	272.03 ± 2.38 ^k^
Šibenik (10)	333.83 ± 2.06 ^a^	2206.79 ± 1.01 ^f^	1971.61 ± 1.88 ^e^	235.18 ± 1.88 ^l^
Šolta (11)	112.33 ± 1.52 ^f^	1987.78 ± 1.88 ^n^	1925.35 ± 0.86 ^f^	62.42 ± 2.43 ^q^
Hvar (12)	210.91 ± 2.03 ^c^	1577.40 ± 2.63 ^q^	1549.04 ± 2.39 ^m^	28.36 ± 2.16 ^r^
Vis (13)	94.33 ± 2.48 ^hi^	2141.29 ± 1.28 ^i^	2061.07 ± 2.29 ^c^	80.22 ± 2.82 ^p^
Korčula (14)	73.34 ± 1.91 ^kl^	2394.96 ± 2.65 ^c^	1976.71 ± 0.95 ^d^	418.24 ± 2.49 ^i^
Mljet I (15)	154.81 ± 2.54 ^e^	2180.59 ± 1.39 ^g^	1968.74 ± 1.85 ^e^	211.86 ± 2.81 ^m^
Mljet II (16)	87.55 ± 2.65 ^ij^	2134.88 ± 1.05 ^j^	1533.91 ± 1.41 ^n^	600.97 ± 1.41 ^e^
Pelješac (17)	101.99 ± 2.25 ^gh^	2693.81 ± 2.84 ^a^	2132.73 ± 2.73 ^a^	561.08 ± 0.09 ^f^
Dubrovnik (18)	108.59 ± 1.68 ^fg^	2588.23 ± 0.93 ^b^	2065.42 ± 2.91 ^b^	522.81 ± 2.91 ^g^
ANOVA	*p* ≤ 0.0001	*p* ≤ 0.0001	*p* ≤ 0.0001	*p* ≤ 0.0001

Different letters (a–f) indicate significant differences between means at *p* ≤ 0.0001. TNFC: total non-flavonoid; TPC: total phenol; TFC: total flavonoid.

**Table 3 molecules-25-05102-t003:** Pigment compounds content of strawberry tree fruit from different locations.

Location	TAC (mg/kg FW)	β-Carotene (µg/100 g FW)	Lycopene (mg/g FW)
Rovinj I (1)	ND ± 0.001 ^h^	84.97 ± 0.01 ^j^	ND ± 0.001 ^d^
Rovinj II (2)	3.08f ± 0.62 ^gh^	134.47 ± 0.001 ^i^	ND ± 0.001 ^d^
Lošinj (3)	4.51f ± 0.94 ^e^	323.87 ± 0.02 ^d^	ND ± 0.001 ^d^
Cres I (4)	4.1f ± 1.42 ^g^	140.42 ± 0.001 ^i^	ND ± 0.001 ^d^
Cres II (5)	8.41 ± 1.98 ^cd^	402.14 ± 0.001 ^b^	0.31 ± 0.001 ^b^
Cres III (6)	ND ± 0.001 ^h^	332.61 ± 0.001 ^d^	1.14 ± 0.001 ^a^
Krk (7)	13.74 ± 2.49 ^b^	205.04 ± 0.001 ^gh^	0.21 ± 0.001 ^c^
Zadar (8)	4.92 ± 0.62 ^fe^	157.76 ± 0.01 ^i^	ND ± 0.001 ^d^
Tisno (9)	11.28 ± 0.36 ^bc^	187.42 ± 0.001 ^h^	ND ± 0.001 ^d^
Šibenik (10)	ND ± 0.001 ^h^	228.73 ± 0.001 ^fg^	ND ± 0.001 ^d^
Šolta (11)	24.19 ± 2.33 ^a^	50.07 ± 0.001 ^k^	ND ± 0.001 ^d^
Hvar (12)	5.54 ± 0.62 ^def^	244.64 ± 0.001 ^f^	0.31 ± 0.001 ^b^
Vis (13)	5.74 ± 0.35 ^def^	ND ± 0.001 ^l^	ND ± 0.001 ^d^
Korčula (14)	ND ± 0.001 ^h^	270.96 ± 0.001 ^e^	0.21 ± 0.001 ^c^
Mljet I (15)	1.23 ± 0.001 ^gh^	ND ± 0.001 ^l^	ND ± 0.001 ^d^
Mljet II (16)	7.59 ± 2.56 ^de^	ND ± 0.001 ^l^	0.21 ± 0.001 ^c^
Pelješac (17)	21.73 ± 1.28 ^a^	369.23 ± 0.01 ^c^	0.31 ± 0.001 ^b^
Dubrovnik (18)	7.38 ± 0.62 ^de^	560.89 ± 0.01 ^a^	ND ± 0.001 ^d^
ANOVA	*p* ≤ 0.0001	*p* ≤ 0.0001	*p* ≤ 0.0001

TAC—total anthocyanin content; ND—not determined. Different letters indicate significant differences between means at *p* ≤ 0.0001.

**Table 4 molecules-25-05102-t004:** Pigment compounds content of strawberry tree leaves from different locations.

Location	Chlor_a (mg/g)	Chlor_b (mg/g)	T_chlor (mg/g)	TCA (mg/g)
Rovinj I (1)	0.75 ± 0.01 ^c^	0.79 ± 0.03 ^b^	1.54 ± 0.04 ^c^	0.18 ± 0.02 ^d^
Rovinj II (2)	0.73 ± 0.01 ^d^	0.56 ± 0.01 ^f^	1.29 ± 0.02 ^f^	0.16 ± 0.006 ^f^
Lošinj (3)	0.44 ± 0.01 ^i^	0.67 ± 0.01 ^d^	1.11 ± 0.02 ^h^	0.06 ± 0.01 ^k^
Cres I (4)	0.36 ± 0.01 ^j^	0.32 ± 0.02 ^i^	0.68 ± 0.01 ^k^	0.12 ± 0.01 ^h^
Cres II (5)	0.64 ± 0.02 ^g^	0.60 ± 0.01 ^e^	1.24 ± 0.01 ^g^	0.16 ± 0.01 ^f^
Cres III (6)	0.19 ± 0.01 ^n^	0.21 ± 0.01 ^l^	0.4 ± 0.02 ^o^	0.07 ± 0.005 ^j^
Krk (7)	0.29l ± 0.005 ^m^	0.31 ± 0.02 ^i^	0.6 ± 0.01 ^l^	0.09 ± 0.01 ^i^
Zadar (8)	1.13 ± 0.02 ^a^	1.24 ± 0.03 ^a^	2.37 ± 0.05 ^a^	0.14 ± 0.01 ^g^
Tisno (9)	0.67 ± 0.01 ^f^	0.68 ± 0.01 ^d^	1.35 ± 0.01 ^e^	0.17 ± 0.01 ^de^
Šibenik (10)	0.31 ± 0.01 ^k^	0.26 ± 0.01 ^k^	0.57 ± 0.05 ^mn^	0.12 ± 0.01 ^h^
Šolta (11)	0.68 ± 0.02 ^f^	0.54 ± 0.01 ^f^	1.22 ± 0.01 ^g^	0.21 ± 0.02 ^b^
Hvar (12)	0.28 ± 0.01 ^m^	0.27 ± 0.005 ^k^	0.55 ± 0.005 ^n^	0.09 ± 0.01 ^i^
Vis (13)	0.7 ± 0.01 ^e^	0.61 ± 0.01 ^e^	1.32 ± 0.01 ^f^	0.21 ± 0.01 ^b^
Korčula (14)	0.93 ± 0.05 ^b^	0.75 ± 0.01 ^c^	1.68 ± 0.01 ^b^	0.27 ± 0.005 ^a^
Mljet I (15)	0.74 ± 0.01 ^d^	0.66 ± 0.01 ^d^	1.4 ± 0.01 ^d^	0.19 ± 0.01 ^c^
Mljet II (16)	0.44 ± 0.01 ^i^	0.39 ± 0.01 ^h^	0.84 ± 0.01 ^j^	0.14 ± 0.01 ^g^
Pelješac (17)	0.29 ± 0.01 ^l^	0.29 ± 0.02 ^j^	0.59 ± 0.02 ^lm^	0.09 ± 0.01 ^i^
Dubrovnik (18)	0.55 ± 0.06 ^h^	0.46 ± 0.01 ^g^	1.01 ± 0.01 ^i^	0.17 ± 0.01 ^e^
ANOVA	*p* ≤ 0.0001	*p* ≤ 0.0001	*p* ≤ 0.0001	*p* ≤ 0.0001

Chlor_a—chlorophyll a; Chlor_b—chlorophyll b; T_chlor—total chlorophyll content; TCA—total carotenoid content. Different letters indicate significant differences between means at *p* ≤ 0.0001.

**Table 5 molecules-25-05102-t005:** Location geographical coordinates, meteorological data [16] and soil type [43] of collected wild strawberry tree populations within the Adriatic coast.

Location Name/No.	Location Coordinates	Area	Average Air Temperatures (°C)	Average Precipitation (mm)	Number of Sunny Days	Soil Type
Rovinj I (1)	45°06′38.4″, 13°36′58.3″	Istria	15	896	98	Rhodic/Chromic Cambisol
Rovinj II (2)	45°03′16.4″, 13°40′34.5″		15	896	98	Rhodic/Chromic Cambisol
$\bar{x}$			15	896	98	
Lošinj (3)	44°35′49.1″, 14°24′35.7″	Kvarner	16.3	1216.5	89	Chromic Cambisols
Cres I (4)	44°58′25.6″, 14°23′13.4″					
			16.3	1216.5	89	Chromic Cambisols
Cres II (5)	44°48′30.2″, 14°21′51.9′					
			16.3	1216.5	89	Chromic Cambisols
Cres III (6)	44°36′53.3″, 14°30′42.1″					
			16.3	1216.5	89	Chromic Cambisols
Krk (7)	45°06′27.1″, 14°39′40.9″					
			16.3	1216.5	89	Chromic Cambisols
$\bar{x}$			16.3	1216.5	89	
Zadar (8)	44°06′43.51″, 15°13′46.34″	Central Dalmatia	16	1139.2	123	Rendzic Leptosol
Tisno (9)	43°48′20.57″, 15°39′10.16″					
			15.9	733.6	122	Rendzic Leptosol
Šibenik (10)	43°45′51.75″, 15°51′20.16″					
			15.9	733.6	122	Rendzic Leptosol
$\bar{x}$			15.93	868.8	122	
Šolta (11)	43°22′45.7″, 16°19′54.7″	South Dalmatia	17.3	706.1	127	Leptic Chromic Cambisols
Hvar (12)	43°10′33.05″, 16°37′5.44″					
			17.3	706.1	127	Leptic Chromic Cambisols
Vis (13)	43°3′10.58″, 16°7′54.10″					
			16.9	726.4	147	Leptic Chromic Cambisols
Korčula (14)	42°55′29.01″, 17°05′10.11″					
			17.3	706.1	127	Leptic Chromic Cambisols
Mljet I (15)	42°43′20.78″, 17°38′53.56″					
			17.3	706.1	127	Leptic Chromic Cambisols
Mljet II (16)	42°45′55.58″, 17°25′17.89″		17.3	706.1	127	Leptic Chromic Cambisols
Pelješac (17)	42°49′22.42″, 17°40′44.07″					
			17.4	844.9	145	Leptic Chromic Cambisols
Dubrovnik (18)	42°39′2.68″, 18°8′26.30″					
			17.4	844.9	145	Leptic Chromic Cambisols
$\bar{x}$			17.3	831.6	134	

$\bar{x}$—mean value.

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
