# Peer review of "Biological Potential of Fruit and Leaves of Strawberry Tree (Arbutus unedo L.) from Croatia"

_molecules, 2020, doi:10.3390/molecules25215102_

Round 1

Reviewer 1 Report

Review’s reply

The paper of Šic Žlabur et al. evaluated the content of chemicals, phytonutrients, antioxidants and radical scavenger capacity of fruit and leaf of wild Strawberry tree populations also considering the influence of micro-location on these parameters.

The reviewer believes that this work is well-organized and sufficiently described. Even if the authors suggest some potential translational applications of their results, the originality / novelty of these findings are not clearly highlighted. This manuscript may give a good contribution to the field but, as is it, the work raises some concerns which will need to be addressed to improve the manuscript's quality. Therefore, the manuscript can be recommended for publication provided the major revisions outlined below have been adequately addressed by its authors.

REVIEWER COMMENTS

Introduction

The introduction could be improved also including some recent references, for instance, the paper below should be quoted to up-to-date the references:

-Tenuta, M.C.; Deguin, B.; Loizzo, M.R.; Dugay, A.; Acquaviva, R.; Malfa, G.A.; Bonesi, M.; Bouzidi, C.; Tundis, R. Contribution of Flavonoids and Iridoids to the Hypoglycaemic, Antioxidant, and Nitric Oxide (NO) Inhibitory Activities of Arbutus unedo L.. Antioxidants 2020, 9, 184.

Some references cannot be readable as they are in Croatian language. Please, the authors should verify Molecules' guideline

Methods

The major concern of this manuscript is the methods are not adequately described. The authors should add a new paragraph carefully describing how they have processed raw materials for getting the extracts of both fruits and leaves for the analysis of bioactive molecules content.

Statistical Analysis

This paragraph should be rewritten. An inter-group statistical analysis among micro-areas should be carried out.

Results

The results related the influence of micro-location are not clearly showed. The tables and the statistical analysis of data should help the reader to identify the cited micro-areas and the significant results. The authors should better arrange the visual and graphical representation of their results to gain clarity.

These sentences are either incomplete or unclear:

Lanes 381-382: “as considered the statistical level of significance by using”.

Lanes 396-397: “Based on the mentioned”. What does it mean?

Lanes 275-276: “a strong dispersion of β-carotene results was determined”

The manuscript is not very easy to read and should be revised by English native.

Reviewer 2 Report

The article corresponding to the ref. molecules-971871, entitled “Profile of Bioactive Compounds in Fruit and Leaf of Strawberry Tree (Arbutus unedo L.)” is aimed at evaluating establishing the value of the fruit and leaves of Arbustus unedo L., as a source of bioactive compounds that could contribute to human health if incorporated as an ingredient in the development of functional foods or nutraceuticals. To the present date, this food matrix has been characterized on their phenolic composition among other bioactive compounds, however, the present work provides a valuable contribution to the field and subject of research by extension to A. unedo by-products, which constitutes an interesting contribution that merits to be explored. Nonetheless, the current version of the article does not meet the scientific standards required to be published in MDPI-molecules and further modifications are required before its acceptance for publication.

Specific comments:

The term phytonutrients used in the abstract is not correct. This could be accepted, to some extent, for the designation of the nutrients present in plant-based foods, but it is not acceptable to include in this term the so-called “phytochemical compounds” that involve non-nutrient secondary metabolites of plants featured by bioactive properties that could provide benefits for human (or mammals in general) health.

In respect to the aims of the article stated in the last paragraph of the introduction, it should be indicated the features of the diverse zones selected for harvesting plant material (distinct climate, representativeness regarding the production of A. unedo,… or whatever reason behind the selection of the diverse geographic areas).

In the section “Results and discussion” and specifically relative to the sub-section “Basic chemical composition of Strawberry tree fruit” It would be introduced the nutritional significance of the diverse parameters evaluated regarding the proximate composition of A. unedo.

On the other side, it is difficult to follow the association of the different locations with the diversity of the soil and weather conditions in Croatia. This info should be provided previously, supported by the presentation of the data in a dedicated Table/Figure. So, somehow, presenting information by this structure would allow obtaining more meaningful results.

This is of special relevance because the plant material used for developing the characterizations described in the present work was collected in a single season (2017). In general, it is well accepted that, especially under the current climate change situation, and given the close relationship between the agro-climatic conditions and the composition of edible plant material and by-products. Hence, the information presented would allow described more rationally the results obtained.

This is of application to the separate sections (2.1 to 2.5) describing the composition of the plant materials analyzed following the structure used by other authors in previous research (e.g., Dominguez-Perles et al 2016 JFCA 53 69 76).

The legend of Figures 1 and 2 do not provide the information required for the correct interpretation. These should be updated by including reference to the Table/Figure showing the information on the diverse geographic areas and climatic conditions.

On page 6, line 199, Table 2 seems to be miss cited. It seems to be referred to Table 1.

The tables included in the current version of the MS should be deeply corrected. Data should be presented as mean ± SD (some values in Tables 2 and 3 do not present SD) and the significance letters (the inclusion of these after the mean does not allow the correct identification of the differences between conditions). All data should be presented in a uniform manner. Some values do not include SD. Correct this.

Nowadays, there is a broad literature regarding the phenolic composition of A. unedo. In this regard, the information provided by the present work does not constitute a significant advance from the current state-of-the-art, especially because of the lack of the chromatographic (identification/quantification) analysis of the individual compounds taking part in the diverse groups of bioactive compounds quantified by spectrophotometric methods. Indeed, this is a methodology useful for the initial check of the value of given plant material as a source of bioactive compounds but does not allow deepening in its actual value, especially as a source of bioactive phytochemicals.

In subsection 2.4, the headline should be redrafted since colored compounds are also important bioactive molecules (as described in the own text of the sub-section), and this aspect is not mirrored by the option chosen by the authors. A tentative title for the sub-section could be “Bioactive colored compounds of…”.

In respect to sub-section 2.5, Since the radical scavenging activity is dependent on compounds described in the diverse Tables (1-3), data referred to this biological activity should be presented in an independent Table/Figure after Table 3.

Regarding materials and methods, in the sub-section 3.5, the text does not describe the application of the multiple range test providing the “significance letters showed in Tables 1-3. In addition, this analysis should be extended to data presented by Figures 1 and 2.

As a final comment, the authors suggest the domestication of this species as a way to obtain a sustainable source of plant material characterized by a valuable content of bioactive phytochemicals; however, frequently, the domestication process has a critical effect on the concentration of such bioactive compounds reducing the value of the materials intended. This fact should be also discussed critically in the proper sections.

Reviewer 3 Report

Manuscript ID: molecules-971871

Title: Profile of Bioactive Compounds in Fruit and Leaf of Strawberry Tree (Arbutus unedo L.)

General comments

The paper analyses physico-chemical composition and content of some bioactive compounds in the leaves and fruits of Strawberry trees harvested from several different locations along the Mediterranean area of the Republic of Croatia.

The basic idea of the manuscript is good, and it could be of practical interest.

However, there are some mistakes and some information is missing

TITLE

I think that the title does not show the content of the manuscript.

Some chemical parameters and some bioactive components were determined, but not the profile of bioactive compounds.

RESULTS AND DISCUSSION

I think that it is not the basic composition that is determined because there are no determinations of ash, fat, protein, ... I think that it is rather a physical-chemical composition

Why are leaves and fruits separated in chemical composition?

It seems that the climatic and meteorological conditions influence the determined parameters. It would be appropriate to collect in a table these conditions in each collection area.

In order to be able to compare them, I think that the physical-chemical composition data would be better in a Table as well as the bioactive components.

In any case, homogenize the format of the figures.

In general, the manuscript was mostly build on the enumeration of the results without a proper discussion in some sections

 MATERIALS AND METHODS

In general, the methods are poorly described. Reference is made to articles that use very different matrices (must, wines, fish, …) so it is necessary to specify the preparation of the samples

REFERENCES

Check last names. For example: Change: “Ruiz-Rodriguez, Fernandez-Ruiz, Sanchez-Mata, Diez-Marques, Tardio” by “Ruiz-Rodríguez, Fernández-Ruiz, Sánchez-Mata,  Díez-Marqués,  M. Pardo-De-Santayana, Tardío”

Reviewer 4 Report

In general:

The paper entitled “Profile of Bioactive Compounds in Fruit and Leaf of Strawberry Tree (Arbutus unedo L.) by Šic Žlabur, Bogdanović, Voća and Skendrović Babojelić reports an interesting study. The topic is very relevant today. In my opinion, this paper can be considered for publication in Molecules. However, manuscript should be improved. 

In detail:

This paper does not bring any new knowledge on profile of bioactive compounds in fruit and leaves of strawberry tree. Additionally, the antioxidant properties of strawberry tree fruits has already been studied. In my opinion more data on identification and characterization of individual phrnolic compounds (phenolic acids, flavonols and anthocyanins) carotenoids, tocopherols of fruit and leaf of wild Strawberry tree populations among the Adriatic coast in Croatia, are needed.

In general, the introduction is well-written and clearly describes the goals of the work. However, it seems us to be necessary to make some minor modifications:

Please include also new references about this part. For example, into the field of the profile of bioactive compounds in fruit and leaves of strawberry tree authors can cite the papers of Agostinho M. R. C. Alexandre et al. Phenolic Compounds Extraction of Arbutus unedo L.: Process Intensification by Microwave Pretreatment. Processes 2020, 8(3), 298; https://doi.org/10.3390/pr8030298 and Hafida Zitouni et al. Phytochemical Components and Bioactivity Assessment among Twelve Strawberry (Arbutus unedo L.) Genotypes Growing in Morocco Using Chemometrics. Foods 2020, 9(10), 1345; https://doi.org/10.3390/foods9101345

The experimental design is well described and the analytical procedures were correctly applied. There were many good discussion and cited references, however, the results is not easy to follow up. The authors provide the results obtained with large detail and sometimes the authors repeat the same concepts so the paper it´s heavy to read.

Round 2

Reviewer 1 Report

Dear Authors,

most of the concerns have been addressed according the review's suggestions. The quality of  manusript was improved as well as the results section, hence the manuscript can be accepted in present form and recommended for publication in Molecules.

Reviewer 2 Report

Thank you very much for your support to complete the revision of the article molecules-971871.

After revision of the MS attached to your mail, I have double-checked the comments of the authors and the modifications done in the reviewed version. So, accordingly, the authors have done the modifications suggested in the MS, and the MS has now the quality requested for its publication in Molecules-MDPI.

Reviewer 3 Report

The majority of the comments have been emended in the revised manuscript.

I think that the manuscript can be accepted in present form

Reviewer 4 Report

The authors have addressed all my concerns and therefore I support publication without further changes.